# Thematic Daily Sleep Routine Analysis of Adults Not in Employment Living with Type 2 Diabetes Mellitus

Rachael M. Kelly [1], John H. McDermott [2] and Andrew N. Coogan [1,*]

1 Department of Psychology, Maynooth University, National University of Ireland, W23 X021 Maynooth, Ireland; rachael.kelly.2013@mumail.ie
2 Academic Department of Endocrinology, Royal College of Surgeons in Ireland, Connolly Hospital Blanchardstown, D15 X40D Dublin, Ireland; johnmcdermott@rcsi.ie
* Correspondence: andrew.coogan@mu.ie

**Abstract:** Background: Day-to-day variations in sleep timing have been associated with poorer glycemic control in type 2 diabetes mellitus, although the factors that influence this sleep timing variability are poorly understood. Methods: Daily routines of sleep in a sample of seventeen adults with type 2 diabetes mellitus who were either retired or not currently working were examined qualitatively through the application of semi-structured interviews and a thematic analysis of the resulting transcripts. Results: Four themes were identified: "Consistent Sleeping Patterns", "Fluctuating Sleep Timing", "Night-Time Disruptions" and "Lasting Effort Needed with Type Two Diabetes Mellitus". The subthemes reflected that many participants had consistent sleep schedules across the seven-day week, but that a desire to maintain a sense of normality, household routines, television schedules and socializing were associated with different sleep timing on weekends. Active disease monitoring and timed medication taking were not identified as important factors in shaping sleep timing. Nocturia, stress and rumination were identified as important factors linked to disrupted sleep. Sleep was not reported as an issue discussed during routine clinical care. Conclusion: Sleep timing in participants appears to be driven by interacting psychosocial and physiological factors, although active disease management does not emerge as a major influence on sleep schedules.

**Keywords:** sleep habits; diabetes mellitus; type 2; qualitative research; retirement

## 1. Introduction

The daily timing of sleep is recognized as an important variable in the relationship between sleep and metabolic health [1]. One manifestation of variation in sleep timing between days is social jetlag, defined as the difference in the timing of midsleep between days with "work" commitments and days without such commitments [2]. Greater social jetlag may be important for metabolic health as it has been associated with obesity and type 2 diabetes risk and severity [3–6]. The original conception of social jetlag as impacting people during their working life [7] is challenged by recent research showing that social jetlag persists to some degree after retirement [8–10]. The factors driving social jetlag in retirement are unclear, as the same working time constraints do not pertain to retirees as for younger age groups; however, it is important to note that social zeitgebers, including social interaction and personal relationships, may influence circadian rhythms and sleep timing [11]. Garefelt and colleagues [8] suggested that such social zeitgebers, including social interactions and TV watching, influence sleep timing, as may "derived social zeitgebers", whereby the schedule of an individual's partner shapes sleep timing. As such, psychosocial perspectives are needed to understand the nature of sleep timing in older adults and its impacts on psychological and physical health. Further, given that both social jetlag and daily sleep timing variability have previously been associated with disease outcomes in type 2 diabetes mellitus [3,12–14], and that type 2 DM is predominantly a disease of middle and older age [15], understanding the social and societal drivers of sleep timing will be of

importance in framing sleep advice around modifiable behaviors for retired individuals with type 2 DM. For example, previous research has shown that individuals with diabetes report that timing of food is impacted by medication [16], although it is not well described what impact this could have on sleep/wake timing.

Qualitative research can provide rich, nuanced, contextualized detail that is not available from the quantitative research that has predominated the study of sleep timing to date and, as such, investigations of lived experiences may help guide future public health policy and inform clinical practice. The aim of this study was to investigate the extent to which social factors shape sleep timing in retired individuals, or individuals who were not currently working, with T2D. The current research questions were (1) Do "social zeitgebers" impact sleep timing in retired/not currently working adults with type 2 DM?; (2) How do such "social zeitgebers" contribute to social jetlag?; (3) What factors influence the consistency of sleep schedules across the week?; (4) Are there any modifiable behaviors commonly associated with social jetlag, or behaviors that are associated with consistent sleep schedules?

## 2. Results

Demographic descriptors of the 17 participants are presented in Table 1 the age range was 48–77 years (mean age: 64 years); 11 participants were retired and 6 were not currently working. Nine participants were male and eight were female, and fourteen participants reported using medication for their diabetes. Ten participants were scored as having no social jetlag from their answers on the Munich Chronotype Questionnaire.

**Table 1.** Descriptive characteristics of the sample.

| Participant | Age | Gender | Work Status | Duration of T2D | Smoker | Household Size | Medication | Insulin | MSFsc (hh:mm) | SJL (h:m) |
|---|---|---|---|---|---|---|---|---|---|---|
| 1 | 60 | M | Retired | 9 months | No | 2 | Yes | No | 4:37 | 0:45 |
| 2 | 55 | M | Currently not working | 1 year, 11 months | No | 3 | Yes | No | 3:43 | 0:23 |
| 3 | 70 | F | Retired | 4 years, 8 months | No | 1 | No | No | 4:05 | 0:00 |
| 4 | 65 | M | Retired | 11 years, 6 months | No | 5 | No | No | 2:40 | 0:00 |
| 5 | 63 | F | Retired | 16 years | No | 2 | Yes | Yes | 5:05 | 0:00 |
| 6 | 73 | M | Retired | 16 years | No | 2 | Yes | No | 2:30 | 0:00 |
| 7 | 57 | F | Currently not working | 17 years | No | 4 | No | Yes | 2:57 | 1:07 |
| 8 | 48 | F | Currently not working | 7 months | Yes | 2 | Yes | No | 3:09 | 0:00 |
| 9 | 61 | F | Currently not working | 1 year | Yes | 1 | Yes | No | 5:13 | 3:15 |
| 10 | 51 | M | Currently not working | 15 years | Yes | 6 | Yes | Yes | 1:53 | 0:23 |
| 11 | 72 | M | Retired | 6 years | No | 2 | Yes | No | 3:45 | 0:00 |
| 12 | 64 | M | Currently not working | 7 years | No | 2 | Yes | No | 3:08 | 0:00 |
| 13 | 73 | F | Retired | 7 years | No | 2 | Yes | No | 2:15 | 0:04 |
| 14 | 66 | F | Retired | 5 years | Yes | 5 | Yes | No | 4:37 | 0:23 |
| 15 | 77 | F | Retired | 10 years | No | 3 | Yes | No | 3:34 | 0:00 |
| 16 | 65 | M | Retired | 16 years | No | 5 | Yes | No | 3:30 | 0:00 |
| 17 | 70 | M | Retired | 22 years | No | 2 | Yes | Yes | 3:15 | 0:00 |

MSFsc: Midsleep on free days (sleep corrected) calculated from the Munich Chronotype Questionnaire; SJL: Absolute social jetlag calculated from the Munich Chronotype Questionnaire.

Four themes were generated from the interview data: "Consistent Sleeping Patterns", "Fluctuating Sleep Timing", "Night-Time Disruptions" and "Lasting Effort Needed with T2DM".

Theme 1: Consistent Sleeping Patterns

*"Well, I sleep in and around the same time every night like."* (Participant 13)

Participants described very consistent sleeping patterns wherein they went to bed and woke up around the same time every day. The four subthemes generated were "habit or routine", "unavoidable morning curtailments", "retirement and increasing age induced similarities" and "ownership over environment before bed".

Subtheme 1: Habit or Routine

A number of participants reported consistent daily routines of sleep/wake: Participant 17 described themselves as a "fairly routine sort of person, you know what I mean, same time everyday kind of". When Participant 16 was asked to talk about any differences

between a typical weekday and a weekend they stated, "There's no difference, it's the same, every day is the same", while Participant 12 remarked, "it's the exact same thing every day". It appeared as if these participants preferred this consistency of daily sleep schedule: "I like to be organised and up in bed by 12" . . . "even though I wouldn't have anything in particular to attend too I still like to have that getting up time" (Participant 3). Participant 13 reported getting up consistently as they enjoy the early morning and that it was a natural time for wakening: "I love getting up in the mornings and I'm a good person for going to bed early at night. It's nine o'clock to bed for me. . .. I wake at quarter to eight and I get up at eight o'clock, yeah. . . I do wake naturally at that time, I'm used of it like. I wake at the same time yeah." Participant 15 described this natural waking time while on holidays: "I mean I am, you know we just went away last week for a few days. . . we woke up at exactly the same time even in the hotel. So, I suppose we're just into that routine now, you know wakening up at around seven-ish". Some participants may have slept consistently throughout their working lives and transitioned naturally into having a very consistent sleeping pattern after retirement: ". . .yeah, I mean, let's be honest, I haven't set an alarm clock-, the only time I've ever set an alarm clock is we got, you know, some transportation to catch, a plane or a train" (Participant 4).

Subtheme 2: Unavoidable Morning Curtailments

Some necessary daily morning curtailments were reported as making it more likely for participants to maintain consistent sleeping patterns. Participant 5 described having to wake to check their blood sugar levels: "just for a sort of safety reasons 9 o'clock, I wouldn't let it go much later than 9, ehh just to check my bloods. . . So I suppose like my blood checkers and medication and that do dictate my time to a certain extent, but not it's not intrusive, it's not that I'd notice it as such". Participants also spoke about how pets can contribute to this consistent schedule; Participant 16 had to rise to care for his dogs: "I get up every morning we'll say around 7, and I'll go out, I have dogs, I let the dogs out". Light entering the room was also identified as a factor contributing to consistent wake times: "I think it's the bright, the brightness is what's waking me". (Participant 3); "I suppose the only other thing is daylight, in that, yeah, I mean I suppose part of what governs when you get up is how light it is in the bedroom when you when you first come to". (Participant 4). Finally, it is also possible that the schedule of their retired partner may influence wake time in the morning. Participant 6 noted that they "work away together", whilst another participant (15) described a situation where her husband's wake time impacted her own wake time: ". . . my husband tends to wake about 7:00, I might sleep a bit later, but when he starts to move I kind of wake up".

Subtheme 3: Age and Retirement Reduce Curtailments

Some participants described how the seven days of the week are similar in terms of activity during retirement: "That's really most days, you know. When you retire the weekends don't matter as much as when you're not retired, you know, so that would kind of be seven days a week, yeah" (Participant 15). Participant 5 similarly said: "Yeah, I mean, as I said normally the seven days of the week would be the same . . .but in the normal world all the days would really be more or less the same." Participant 4 commented on how these retirement-related changes in daily schedule may be a challenge: ". . . they're quite similar across the week now. . . .in fact one of the challenges I think when you pack up work is to try and make not every day the same".

Subtheme 4: Ownership over Environment

A number of participants described an awareness of how various habits and rituals impacted sleep timing. For example, Participant 15 noted: "I find it virtually impossible to go asleep unless I read for a while. . . have to read, I mean I absolutely have to read. I remember we went to a hotel one time, and I forgot my book and I read all the brochures. I literally have to read something before I go asleep". The role of watching television and device usage in determining sleep habits was noted by participants: "We tend not to watch TV after about ten, ten thrity-ish and at the moment we, we're watching prison break and

there's a lot of adrenaline in that". (Participant 5). "I find that if I spend time on my tablet looking up emails or newspapers cause I get the newspapers online, that would keep me awake if I was on it before I go to bed. So, I have to make sure I'm not doing that, I try and get that done during the day". (Participant 3).

Theme 2: Fluctuating Sleep Timing

> *"I don't know it just- I'm not really somebody who has, uhm, regular habits, so I just sort of go with the flow."* (Participant 8)

Some participants reported varied sleep timing day to day and flexibility around their sleep timing as they were not working. This might be due to factors such as getting up earlier some mornings to walk or relaxing for longer some evenings before bed. Three subthemes were generated: "Quality TV", "Maintaining a Sense of Normality", and "Derived Zeitgebers".

Subtheme 1: Quality of TV

For some participants, daily variations in sleeping patterns were reportedly driven by the perceived quality of TV programs available on a given night: "I go maybe sometimes maybe half eleven, more times maybe twelve/half twelve... Now there might be an odd night I might go late I might go d'ya know if there's something on it might be one o'clock/half 1 but not always, its mostly around the twelve-ish, eleven thirty, twelve or it could be half twelve" (Participant 11). The bidirectional relationship between television watching and sleep was also noted: "Nothing in particular but you know I'd say well it's time to go now, (wife's name) might be gone before me you know. I just go because there's nothing much on TV and I-I'll go, you know... depends how tired I'm starting to feel, depends on what I want to watch on TV, depends on if I'm talking to my daughter, you know. I don't know it just- I'm not really somebody who has regular habits, so I just sort of go with the flow... well this is gonna be strange cause I kind of I go to my room and I watch TV for quite a while and I might try and fall asleep about half ten, but if I can't get sleep I'll watch TV again" (Participant 8). Participant 16, who reported a very consistent sleep schedule seven days a week, stated that, occasionally, if he started watching something interesting on the TV he might sleep later: "sometimes I might go a bit later, depends if there's something on tele interesting".

Subtheme 2: Maintaining a Sense of Normality

Participants reported maintaining consistent schedules during the "working" week but staying up later and getting up later at the weekend in an effort to maintain some distinguishing features (and many of these participants displayed social jetlag from their questionnaires; Table 1); "I do want to keep the weekend different, because every day just becomes the same... I go to bed later on a Friday night, I'm watching YouTube and stuff. And knowing that, I still want to get my eight hours then I don't wake up till about nine. It's not something I have to sort of wake up at eight and say, oh I'm staying till nine, I don't wake up till about nine on Saturday anyway. Yeah, and certainly Sunday" (Participant 1). Participant 2 described a similar situation wherein he consistently slept and woke around the same time during the week due to household factors and medication, but delayed at the weekend to socialize with his family: "Maybe a weekend now, Saturday night we might be a bit later up because we might as a family, we might be sitting around and watch a movie or watch something maybe we missed during the week or had recorded during the week. My wife takes a glass of wine or whatever maybe on a Saturday night so I would have a diet drink or whatever, my daughter might have a glass of wine or whatever. So, and we might be up until eleven or maybe half eleven at the weekend, you know what I mean". The same participant noted that the need to take medication limited the amount of time he would sleep in for on weekend mornings: "Well again its around medication really. Because one of the tablets I have to take roughly about a half an hour to an hour before I take any food. So that would influence me, taking that tablet first thing. So at weekends, well Saturday, that would be no later than maybe eight o'clock I would take that tablet, so I could have my breakfast at nine. Sunday, maybe, a wee bit later I might take it maybe half eight and then having me breakfast at half nine, but no later than that".

Participant 10 also noted the importance to them of maintaining a weekend/weekday difference for a perceived sense of normality: "...about half ten or eleven o'clock, I'd go to bed depending if there's anything decent on like a movie, that's it, d'ya know. So I would go to bed a little bit later at the weekends so about half ten, eleven. . . for me it was just a sense of normality, being a normal person that like y'know stay up watching like a bit a tele y'know, where cause I don't drink anymore since I was diagnosed with diabetes so just a sense of normality more than anything". Participant 9, who lives alone, noted a similar importance of maintaining a weekend/weekday difference for a sense of normality: "Well I'm often later going to bed on a Friday night, I'd watch a late film, d'ya know. Could be one o'clock, could be two o'clock. And kinda Saturday and Sunday then I'd be back to my week time schedule, d'ya know around eleven".

Some participants noted that they did not move their wake time much at the weekend but did allow themselves more rest: "Well, I would wake up Rachael but I would often go back to bed after going to the bathroom, on a Saturday and Sunday. D'ya know? I wouldn't be up at say six o'clock or seven o'clock I'd go back to bed. I'd be up around ten now on a Saturday or Sunday. Very rarely would I sleep like, I'd be listening to the radio or something d'ya know". . . "I suppose cause d'ya know it's like what would you say, it's like indoctrination hahaha d'ya know. You rest a bit more or whatever or you think you are anyhow at the weekend d'ya know" (Participant 9). Participant 13, who had a consistent sleep schedule throughout the week, also reported resting for a little longer on a Sunday morning: "Ahh no reasons I'm afraid, I just take it easy at the weekend". Participant 15 described very routine sleep Monday to Sunday but did report staying in bed for longer at the weekend: "The only thing that we might do because we got the paper delivered Saturday and Sunday, we might stay in bed a bit later we might not get up 'til nine. But we wouldn't be asleep 'til nine, you know what I mean, we would wake probably the same time but we might sort of stay in bed a bit longer, read the paper, but we would not be sleeping later".

Subtheme 3: Derived Zeitgebers

Daily sleeping patterns were also reportedly influenced by family or other household members' schedules, with some participants reporting not having the opportunity to sleep until their preferred wake time on weekdays due to noise in the house in the mornings: "My wife and daughter would be getting up, my wife would be leaving here at half seven. My daughters up then at that stage and she's getting ready to leave for quarter past eight so you might be snoozing but you're not sleeping-sleeping because y'know what I mean, . . . We go to bed because my wife is up so early, we do go to bed fairly early. Say she would be in bed by half 9 and I would probably be in bed by half 9 but I would tend to watch a wee bit of TV or whatever. So I might watch that for an hour until half 10 and then I would usually switch it off and go to sleep myself after that" (Participant 2). Participant 7 also described how movement in the house during the week can also cause her to wake and how the absence of such disturbances allows her to sleep longer: "I'd usually be awake around six because my husband and son would be getting up and he has to get dropped to work. . . Yeah, if no one is stirring in the house now I might sleep- I might wake up at eight or half eight. . . If there wasn't activity in the house you know I would sleep on a little longer yeah yeah".

Other responsibilities were also reported to influence wake time during the week. For example, Participant 8 had to bring her daughter to school, so she needed to use an alarm clock: "Well, in the week if I'm taking my daughter to school, we'll wake up about seven, amm at the weekend, see ah (*sigh*) I'm kind of strange I wake up early in the morning. I can wake up about half five to go to toilet and if I go back to sleep, if it's the weekend I'm not gonna wake up again about eight". Participant 14 has a son who works shifts and this led to her varying her sleep onset as she would always wait for him to return home before going to bed: "Well my son finishes work at half 12 at night and when he comes home, shortly after that, I go to bed. I like to see him before I go to bed". . . "It could be 12 o'clock on a Friday night because (son's name) comes home earlier on Friday night."

Participant 3, who expressed a strong preference for a consistent sleep schedule, described how if people or family members are visiting this would occasionally disrupt her sleeping pattern: "If you're on your own, it's easier to make those make those decisions for yourself". Participant 11 and his wife slept freely during the week but reported using an alarm clock every Sunday morning for mass: "If I'm going to be up for 8:30 mass, if I'm getting up at seven I try to get to bed around 12 o'clock and have seven hours".

Theme 3: Disrupted Night-Time Sleep

> *"Well I could be getting up during the night, and I mightn't sleep for a while. My sleep wouldn't be great."* (Participant 14)

A number of participants described difficulties initiating sleep and/or wakening after sleep onset: "I'm a poor sleeper really. If I get to bed, if I don't get to sleep fairly quickly, sleep does pass me by, and I could be hours before I sleep. But I don't know what it is but if I go in, lie down and am gone in a few minutes that's fine." (Participant 7); "Usually go back to sleep again pretty quickly, but if it's very close to morning say five or half five sometimes, I do find that hard to get back asleep again" (Participant 15); "I'd say half three in the morning and it was up then for an hour and a bit, made tea and went back to bed again. And at half ten about I got up then" (Participant 14).

Subtheme 1: Rumination

When participants were asked about how long it takes them to sleep or how easily they sleep, it became clear that some participants experience variations in sleep onset linked to rumination and stress: "...it can take a while, like you know, depends again like y'know how you feel like but sometimes you might be a while, like you might be thinking and that type of thing" (Participant 11). Participant 8 also described her problems with sleep due to having things on her mind: "...and I mean, I'm- I am quite anxious person as well, so I think having things on my mind all the time probably doesn't help, so I can't switch off I suppose". Stress and anxiety were also reported as negative influences on sleep: "...I mean there are, there are some stresses, family wise, am which I have... And yeah, so things like that, yeah....so I mean, so those sort of those stresses do- would affect sleep definitely" (Participant 4); "I think you could be disturbed by the newspaper especially with what's going on around this and I think I have to make sure I'm not reading negative- negative news before I go to bed. I think that could impact you a lot" (Participant 3).

Subtheme 2: Waking During the Night to Use the Bathroom

A number of participants described waking during the night to use the bathroom; nocturia is a very common symptom of diabetes and appeared to be prevalent in the current sample: "No matter what and as I say most of my friends who are my age group all say they get up at night, so I think it's kind of that interrupted sleep goes with getting older. So, you know, so I do have broken sleep every night, but I mean I don't know whether this diabetes is causing it, I've never actually talked to the doctor about that. I just get up to go to the toilet so so I don't ever sleep right through from, say, midnight to six or seven without waking-, without getting up at least once" (Participant 15). Participant 16 described a similar pattern that he could control to some extent by reducing fluid intake in the evening: "And if I was lucky enough that I don't have too many drinks of tea, or water, or milk or something, I won't get up as often, but I still get up". While nocturia impacted some participants all of the time, some participants did not report this problem at all, while some experienced it occasionally: "Well maybe an odd time I might have to go to the toilet during the night, that's all. That's the only reason" (Participant 13); "Now sometimes I have to go to the bathroom as well like d'ya know so. The bathroom has an influence I suppose" (Participant 9).

Subtheme 3: Other Health Issues and Age

Other health issues and conditions were also reported to be linked to disrupted sleep. Participant 15 described how her arthritis may further disrupt her sleep beyond the bathroom use: "Well, the-the only thing that would influence me would be that sometimes if I'm having

a bad run with the arthritis I might wake up during the night with my knee, like my knees might be paining me and I take two paracetamol and go back asleep again. But sometimes when I say I'm up twice a night yeah...but if I was going through, you know, a flare up with the arthritis I might wake a third time and that would be just my knees would be paining me". One participant was experiencing menopause, and this disrupted her sleep: "Yeah, I would often wake up every hour. I'm menopausal, hah you know that kinda thing ha I'd be very hot you know that sort of thing. I don't sleep right through the night ever; I wake a good bit. But I guess that's kinda you know the way it you know the way. It's not that once I go to sleep, I'm asleep for the whole night, I'm not, you know" (Participant 7). Participant 6 described the disruptive impact of neuropathy on sleep: "My left leg is affected by the diabetes, I wouldn't sleep one wink if I didn't take the tablets".

### Theme 4: Lasting Effort Needed with T2DM

> *"But it's the one thing you can't do in diabetes is take a break from anything. You have this every day, all day and there's no break from it."* (Participant 7)

Many participants made reference to the constant awareness around their diabetes, noting how serious the disease was and that "it's a lifestyle really" (Participant 7). Three subthemes were identified: "the daily burden of diabetes", "lifestyle and eating adjustments" and "takes an active role in learning about diabetes".

### Subtheme 1: The Daily Burden of Diabetes

A number of participants described the daily efforts and variations inherent in managing their disease: "D'ya know there's variations there's up and down, you're going to experience that as you go along like you know, know as you try to live with your type 2". (Participant 11); "... it's there all of the time. There's no break from it or anything, you just have to do with it every day you know" (Participant 7). The challenges of self-management of type 2 DM were also noted: "I'll tell you what, a lot of it is down to myself and the doctors are leaving it to me, to do myself. They're not, they'll tell you that this is high, that's high, they're not telling you how to go about doing things. You need somebody that's able to tell you, you know, the best way of doing things, to look after things the right way, you know what I mean... if you've somebody who's trying to help you and you know they're trying to help you, you'll try yourself"(Participant 16).

### Subtheme 2: Lifestyle Adjustments

Many participants spoke about the lifestyle changes they had made or were in the process of making in order to optimize their health. Participant 1 took the opportunity after retiring from work to increase his activity levels: "Well, exercise is another fairly important thing, but I was I have like I was always attempting to do 10,000 steps a day even when I was in work". Participant 3 described how she was still managing to keep her diabetes under control with a healthy diet and exercise: "I can't remember what it was and she said that's usually when you need medication and I said, well, I'll try to improve my diet and my exercise and she let me do that so. I did le-, I lost a stone and it came down a bit".

Participants were conscious of what types of food they consumed and how that might influence their diabetes management and health overall: "Well, look it, I am conscious of it, well now I say conscious, I'd be, it would be always sort of there like when I go to sit down to eat or anything like that" (Participant 12). Breakfast consumption seemed to be an important aspect of maintaining a healthy diet for this group and only two participants reported ever skipping breakfast. Participant 14 reported "sometimes" skipping the meal while Participant 16 sometimes went without breakfast which actually led to overconsumption later in the day: "Sometimes I go without breakfast, just with the rushing and then you'd eat too much you know what I mean". Other participants were very enthusiastic about breakfast: "We would eat breakfast, we're big breakfast eaters both of us, so we have quite a substantial breakfast at about nine or half nine". (Participant 15); "Yes, my breakfast never changes, never ever changes... That's every morning, seven days a week, 365 days a year. That's final, I never miss breakfast" (Participant 12).

Some participants were also conscious of not eating too late at night and identified how late-night eating could prevent them from achieving a healthier weight: "…five o'clock is the latest for me. I don't eat any later as I said a while ago that I might just have a rich tea biscuit and a glass of water maybe or maybe if I was really hungry…" (Participant 13). "Well, I I'm aware of the fact that eating heavy food late at night is not good, so we do tend to avoid that." (Participant 8). Some participants reported eating snacks closer to bed and even though they were aware of the negative impact this could have: "I'll tell you, probably too late before I go to bed, that'll be the problem. Probably too late before bed, you see" (Participant 16).

Subtheme 3: Taking an Active Role in Learning About T2DM

Participants reported an interest in learning about the disease beyond what their clinicians told them: "I didn't know nothing about diabetes until I got it, but I tell you I've a lot learned since I got it" (Participant 6). Some participants reported keeping up with research online: "It takes the health board a long time for their nutritionists or their dieticians to change their what you call, thinking, cause they're trained a certain way to believe a certain thing and they're too rigid in their beliefs… And I felt that when I reduced carbohydrates it had a big impact on my body and my sleeping, so I think I might try that again" (Participant 3). Participant 1 noted that he has learned more online beyond what the diabetes team has told him: "They don't really mention sleep, it's only looking at stuff on the internet where you see sleep is also very important".

## 3. Discussion

In this study, we used semi-structured interviews and thematic analysis to investigate drivers of sleep timing and daily routines in participants with a diagnosis of type 2 DM without a regular work schedule. Our motivation was to uncover new insight into factors that contribute to variability in sleep/wake timing in people with type 2 DM which may in turn contribute to poorer glycemic control [3,13]. A greater regularity of sleep/wake times may have beneficial effects through promoting synchrony between night-time sleep episodes, physiological drive for sleep and the internal circadian rhythms [17]. Given that the circadian clock is a key factor in metabolic regulation, factors that promote circadian synchrony may have metabolic benefits [1]. Further, regular bedtimes and wake times in older adults have been associated with better sleep quality [18], which itself is associated with better disease outcomes in Type 2 DM [19]. There has been significant quantitative work undertaken to date that describes the key role of work arrangements in shaping sleep timing variability between work days and work-free days [17,20]. However, there is little understanding of the drivers of sleep timing variations in adults who are not currently working, attending school or who have other explicit obligations that would obviously impact on the timing of their sleep/wake cycles.

Some participants who reported consistent sleep schedules had consciously developed such a routine as a habit; habit formation has previously been suggested as an approach to the adoption of healthy lifestyle decisions [21]. Consistent sleep habits were in part influenced by individual preference for daily organization, even if tasks did not have to be attended to at specific times. Conversely, retirement and older age were identified as enhancing sleep regularity by reducing daily differences, and such a finding is consistent with previous research showing that social jetlag and sleep timing differences between the week and the weekend decrease with age and retirement, largely due to the end of structured employment [8]. Participants with consistent daily patterns also often reported methods of enhancing their environment in preparation for sleep, along the principles of sleep hygiene [22]; these participants reported turning the television off at a pre-determined time, not staying up for no particular reason, not having any screen stimulation in the bedroom and reducing time on the computer or tablet in the hours before bed. Television viewing and computer/screen usage before bed may be biologically significant as blue-wavelength light emitted from these devices may suppress melatonin synthesis and delay sleep onset [8,23]. As previous work has indicated that sleep hygiene is useful in the

management of chronic conditions such as type 2 DM [24], the current results indicate that increased psychoeducation around good sleep habits may be useful in disease management, especially as a number of participants noted that sleep was not discussed during their routine clinical care.

In the absence of the need to wake for work, we were interested in understanding the role medication and active disease monitoring may play in sleep timing. Only one participant reported needing to wake in order to check blood sugars, and two other participants reported being conscious of taking medication at a certain time before food. Medication use had been shown to previously guide food timing [16], but less is known about how it may shape sleep/wake schedules. The majority of participants took their medication an hour before breakfast or with breakfast, but this did not dictate the timing of breakfast or wakening (although there was a high level of breakfast eating in the sample). This finding suggests that the need for timed medication taking and active disease monitoring may not be major influences on sleep timing in type 2 DM. However, it should be noted that detailed information of medication use by participants was not gathered (e.g., type, dosage), and some participants may not have had the requirement to test their blood glucose in the morning as part of their disease management regime. As such, the findings from the current participants may not generalize to other groups.

In examining the factors that shape weekday/weekend differences in sleep timing, the importance of a weekend "treat", a desire to have not all days be the same and the importance of maintaining a weekday/weekend difference for a sense of "normality" emerged as motivations for maintaining weekend sleep timing as different to the rest of the week. The importance of later weekend bedtimes to facilitate socializing with family members and friends was also noted. Whether the physiological benefits of maintaining sleep timing consistency outweigh the psychological benefits is an open question, as is whether a sense of weekday/weekend normality can be maintained alongside sleep timing consistency. In the context of the broader family situation, individuals who are not working do not have direct rise-time constraints stemming from work or school [18], but may be influenced by the derived zeitgebers of the schedule of household members during the working week [8]. For example, when individuals retire but their partners remain in full-time employment, smaller reduction in sleep timing differences are observed in comparison to those with retired partners [8]. These findings may indicate that sleep psychoeducation should be extended to include family members, rather than being directed at just the patients themselves.

In terms of sleep disturbances, many participants reported rarely sleeping through the night and described being "up and down" a lot. Such sleep disruptions become more prevalent with increasing age [18,25] and may be contributed to by both reduced homeostatic sleep pressure and decreased circadian rhythm amplitude [26]. Another common factor that emerged was nocturia requiring waking at least once during the night to void [27]. While nocturia increases with age, it can also be influenced by other factors including obesity and diabetes [28]. Many of the participants reported acceptance of nocturia as part of the aging process, a finding previously reported [28]. There has been recent evidence suggesting that nocturia is indicative of circadian rhythm dysfunction wherein disrupted circadian rhythms cause voids or vice versa [29]. These results reinforce the potential to increase sleep health through successful behavioral and medical management of nocturia.

The lasting effort required for managing diabetes was identified as an important contributor to sleep timing. Participants identified the daily burden of self-management and its influences on daily routines [30]; such efforts and the attendant mental stress and cognitive ruminations may negatively impact sleep timing and quality. Strikingly, participants demonstrated a strong willingness to make lifestyle changes to manage their diabetes, and some participants had researched the role of sleep in type 2 DM management; these findings suggest that structured sleep psychoeducation interventions for adults with type 2 DM could be well received and adhered to.

*Strengths and Limitations*

This study has a number of strengths. Firstly, the semi-structured interviews allowed for rich information to be identified regarding daily schedules. The reflexive thematic analysis utilized allowed for a data-driven approach and for us to give participants a voice. All interviews were conducted online or over the phone due to the restrictions of the COVID-19 pandemic; previous research has identified the benefits of phone interviews in comparison to face-to-face interviews for sharing personal details in that participants may have been more comfortable talking about their daily patterns over the phone or the computer than in person [31]. Furthermore, a recent study investigating the enablers and barriers with self-management of both type 1 and type 2 diabetes obtained rich data from phone interviews [32] which suggests it may be an appropriate method for this particular population and helps reach people living in remote locations.

Weaknesses of the current study include the possible conflation between being retired and older age, and the inclusion of some younger participants who were not working. As sleep timing is profoundly influenced by normal healthy aging [26], it is not immediately apparent that the effects of retirement can be delineated from the effects of increasing age. Secondly, detailed clinical information was not gathered, and so specific factors such as medical or psychological co-morbidities were not examined; it is possible that the sleep patterns of patients with serious complications of type 2 DM would vary from those with well-controlled or uncomplicated disease or with longstanding disease compared to more recently diagnosed disease. Further, the study participants may represent a self-selecting group of patients who are highly motivated and interested in their DM self-management and, as such, results from this group may not generalize well to other type 2 DM patients.

## 4. Materials and Methods

### 4.1. Participants and Recruitment

The study sample was recruited through purposive sampling, with participants fulfilling the inclusion criteria of being 40 years or older, having a diagnosis of T2DM and being either retired or not working at the time of the interview (April–September 2021). Exclusion criteria were currently being in employment, a current diagnosis of sleep apnea or another sleep disorder and currently taking medication for sleep. The study was advertised online through the advocacy groups Diabetes Ireland and Diabetes UK and was also shared online using social media platforms. Specific T2D support groups were also approached and flyers for the study were shared with some diabetes clinics to reach further participants. Beyond this, participants were recruited through word of mouth. Seventeen participants with T2D were recruited as part of this study; recruitment ceased after seventeen participants as initial analysis suggested that the data was high in information power [33]. There was no financial incentive for participation and ethical approval was obtained from the Maynooth University Social Research Ethics Subcommittee (SRESC) before commencing the study.

### 4.2. Data Collection

Data were collected through semi-structured interviews which were informed by a schedule focusing on the reasons for sleep timing choices across the week, with associated meal timing choices and typical activities on a daily basis also discussed (Table 2). Interviews were conducted by RK and lasted between 18 and 33 min (average duration of 24 min) and all interviews took place either online via Microsoft Teams or over the phone, depending on the participants' preferences and access to technology. The interviews were transcribed to produce orthographic transcripts whereby a verbatim script was developed that included all utterances. Demographic details including age, sex, employment status (retired or not currently working), employment status of partner, diabetes duration, household size, medication use and insulin use were also obtained from participants. Participants also completed the Munich Chronotype Questionnaire [34] to allow for quantitative estimation of sleep timing and social jetlag.

**Table 2.** Semi-structured interview questions.

| |
|---|
| 1. What time do you usually wake during the week and what influences this wake time Monday to Friday? Can you tell me a little more about this? |
| 2. Talk me through a typical weekday day. What activities do you do during the day and when? |
| 3. What time do you usually go to bed on a weekday, so the nights preceding Monday to Friday, and what factors affect what time you go to bed at? |
| 4. Would you say you can freely choose your sleep and wake timing on a weekday? Can you say a little more about this? Are there any influences like children/pets/grandchildren/hobbies? |
| 5. What time do you normally wake on Saturday and Sunday, and what factors influence this? |
| 6. Could you please talk me through a typical weekend day? What activities do you do during the day and when? |
| 7. What time do you go to bed at the weekends, so Friday night and Saturday night, and what factors affect what time you go to bed at? |
| 8. Would you say you can freely choose your sleep and wake timing on a weekend day? Can you say a little more about this? Are there any influences like children/pets/grandchildren/hobbies? |
| 9. V1: Talk to me a little bit about how your sleep onset and end are slightly later/earlier at the weekend. Is there any reason for this? (i.e., do you prefer staying up later or getting up later?) Are these curtailments important or unavoidable? Would you be able to keep a consistent schedule across the week? Would you be willing to modify them if it could have a positive influence on your health? V2: Reflecting on your sleep timing behaviors, it seems that you sleep and wake pretty consistently across the week. Are there any other factors not discussed above which might influence this? Do you make a conscious decision to keep this consistency across the week? |
| 10. Is there anything else you would like to add about your week and weekend day sleep timing? |
| 11. Would you often nap during the week or at the weekend? |
| 12. Have you ever tried monitoring or changing your sleeping habits to lose weight or improve your health or for other reasons? |
| 13. Have you ever spoken to your GP/Consultant about your sleep timing or your sleep in general? |
| 14. Do you have any stressors in your life currently that have impacted your sleep timing or sleep in general? |
| 15. In general, would you describe yourself as more of a morning lark or night owl? |
| 16. Do you typically eat breakfast? Can you tell me what time you eat this at, and does it vary between week and weekend days? |
| 17. What do you typically eat for breakfast and does this vary between week and weekend days? |
| 18. Can you tell me a little bit about the timing and content of your other meals on a typical day? |
| 19. Have you ever tried changing your food timing or habits in general to lose weight, improve health or for any other reasons? Please tell me more about this. |
| 20. Has any of this changed with the COVID-19 pandemic? |
| 21. Is there anything else you would like me to know? |

### 4.3. Data Analysis

Reflexive thematic analysis (TA) was used to analyze the data, following the six steps outlined by Braun and Clarke [35,36]. This approach acknowledges the active role that the researcher plays in theme generation and was chosen as appropriate for the investigation of the participant's views and day-to-day experiences of sleep/wake patterns as it allows for data-driven analysis and is suitably theoretically flexible. For coding and analysis, researchers used an inductive framework with semantic-level coding where the focus on meaning was grounded in the data and the explicit statements of the participants. MAX-QDA software (v2020, Verbi Software, Germany) was used to facilitate the familiarization and coding of the data as well as the generation and review of themes, and was also used to help compile quotes that supported the themes.

RK conducted the interviews, transcription and analysis of the data. Familiarization was through data collection and immersion through transcription of all of the interviews. Following accuracy checking, transcripts were imported into MAXQDA and re-read by RK, with initial notes being made. The data were then coded; codes identified were mainly semantic in nature rather than latent, and the analysis is descriptive and summative where the participants were given a voice. After the initial codes were developed and discussed with AC, themes were generated from recurring codes utilizing the creative coding function on MAX-QDA, allowing the researcher to visualize and organize codes. After the initial list of themes was generated, it was reviewed and refined by RK until the coded extracts were adequately captured by the themes. The final themes were then defined and named in order to allow accurate explanation. The final step was the analysis and write-up of the report explaining

how the chosen themes were relevant to the data and the research questions. As this method of TA is not a linear process, the stages were revisited several times to ensure themes were accurate and results were as reflective of the data as possible. All authors, consisting of a PhD researcher in psychology and sleep science (RK), a professor of sleep science (AC) and a consultant clinical endocrinologist (JMcD), reviewed the themes and subthemes for conceptual coherence, credibility and clinical relevance. The researchers acknowledge their involvement in the research field and the potential influence this may had.

## 5. Conclusions

Qualitative research approaches in sleep medicine have strong potential to reveal rich detail on psychological and behavioral influences on sleep habits. The current study reveals the importance of family context, the need for a sense of "normality", the presence of nocturia and the potential for sleep psychoeducation in improving sleep timing consistency as a lifestyle modification for disease management in patients with type 2 DM. We propose that the current findings could be used to guide future quantitative and qualitative research investigating behavioral and psychosocial drivers of sleep timing variability in patients with type 2 DM and inform psychoeducational interventions addressing such sleep timing variability with a view to improving disease management and quality of life.

**Author Contributions:** Conceptualization, R.M.K., J.H.M. and A.N.C.; methodology, R.M.K. and A.N.C.; formal analysis, R.M.K. and A.N.C.; investigation, R.M.K. and A.N.C.; data curation, R.M.K.; writing—original draft preparation, R.M.K., J.H.M. and A.N.C.; writing—review and editing, R.M.K., J.H.M. and A.N.C.; supervision, J.H.M. and A.N.C. All authors have read and agreed to the published version of the manuscript.

**Funding:** This research received no external funding.

**Institutional Review Board Statement:** The study was conducted in accordance with the Declaration of Helsinki and approved by the Research Ethics Committee of Maynooth University (April 2021).

**Informed Consent Statement:** Informed consent was obtained from all subjects involved in the study.

**Acknowledgments:** We would like to acknowledge the assistance of Diabetes Ireland and Diabetes UK for facilitating the study, and thank the research participants for giving so generously of their time.

**Conflicts of Interest:** The authors declare no conflicts of interest.

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
