# Peer review of "Thematic Daily Sleep Routine Analysis of Adults Not in Employment Living with Type 2 Diabetes Mellitus"

_2624-5175, doi:10.3390/clockssleep6010002_

Round 1

Reviewer 1 Report

Comments and Suggestions for Authors

This work by Kelly and colleagues provides a snapshot of the sleeping behaviors of patients with T2D who are also unemployed. The information was collected using a questionnaire/ interview. In general, the article is interesting; it gathers testimonial information and allows the reader to put into perspective how psychosocial factors may play a role in sleeping patterns. Unfortunately, as they highlight in the limitations section; there was no clinical data collected associated with T2D or sleep data describing patients’ sleep patterns. Nonetheless, this work may be considered for publication after addressing the following comments:

1. Please consider improving title: Instead of writing “not in employment” the word “unemployed” could be used. 

2. Please proofread the entire manuscript, there are areas that would benefit from professional English editing. 

3. The structure of the manuscript needs to be reorganized, the methods section should go after the introduction not after the conclusions.

4. Methods: What exclusion criteria did you establish for your study?

5. Could you please specify who conducted the interviews?

Comments on the Quality of English Language

There are areas of the manuscript that requiere English editing. Review sentence construction. 

Author Response

This work by Kelly and colleagues provides a snapshot of the sleeping behaviors of patients with T2D who are also unemployed. The information was collected using a questionnaire/ interview. In general, the article is interesting; it gathers testimonial information and allows the reader to put into perspective how psychosocial factors may play a role in sleeping patterns. Unfortunately, as they highlight in the limitations section; there was no clinical data collected associated with T2D or sleep data describing patients’ sleep patterns. Nonetheless, this work may be considered for publication after addressing the following comments.

  1. Please consider improving title: Instead of writing “not in employment” the word “unemployed” could be used.

Authors’ Reply: We have revised the title to “Thematic Analysis of Daily Sleep Routines in Adults Not in Employment Living with Type 2 Diabetes Mellitus” to better reflect the methodological approach adopted in the study. We have kept “not in employment” as we believe that the common meaning of “unemployed” is those of working age not in employment, and as the study sample included those who were unemployed as well as those who were retired, we feel this is a more accurate phrasing.

  1. Please proofread the entire manuscript, there are areas that would benefit from professional English editing.

Author’s Reply: We have conducted a careful proof-reading of the manuscript, and have highlighted throughout where we have altered phrasing. We also wish to note that participants’ quotes are relayed as spoken, sometimes using Hiberno-English vernacular (eg. “I do be…”).

  1. The structure of the manuscript needs to be reorganized, the methods section should go after the introduction not after the conclusions.

Authors’ Reply: The structure of the manuscript is as laid out in the instructions for authors for this journal.

  1. Methods: What exclusion criteria did you establish for your study?

Authors’ Reply: We have now stated the exclusion criteria in the methods: Line 495-497 “Exclusion criteria were currently being in employment, a current diagnosis of sleep apnea or another sleep disorder and currently taking medication for sleep.”

  1. Could you please specify who conducted the interviews?

Authors’ Reply: We have now clarified that the first author conducted all of the interviews: Line 5011-512 “Interviews were conducted by RK and….”

Reviewer 2 Report

Comments and Suggestions for Authors

Dear authors and editor,

The manuscript titled "A Qualitative Analysis of Daily Sleep Routines in Adults Not in Employment Living with Type 2 Diabetes Mellitus. " aimed to investigate the extent to which social factors shape sleep timing in retired individuals, or individuals who were not currently working, with T2D.

There are some minor and major considerations that I would like to discuss with the authors.

Abstract

1-Title:Adequate, the authors identify the design and subject matter of the study.

2- Adequate: The abstract gathers the relevant information of the manuscript.

3-It is recommended not to include abbreviations in the abstract without prior explanation.

"Results: Four themes were identified: ”Consistent Sleeping Patterns”, “Fluctuating Sleep Timing”, “Night-Time Disruptions” and “Lasting Effort Needed with T2DM”. 

4-Change the keywords. Delete the words "Thematic Analysis", "Sleep timing"; "Diabetes"; and "Qualitative". Not found in the MeSH (Medical Subject Headings). It is recommended to put the keywords as they appear in the pubmed thesauri.

Introduction

5-Adequate: The most important concepts of the subject to be developed are identified.

Materials and Methods  

6-Why do the authors incorporate the material and method at the end of the manuscript?

Results

7-Adequate. The results are correct. The authors present the findings of the study in an organized manner..

Discussion

8-The authors are aware of the limitations of their study design.Could the disparity in disease debuts be a limitation?

Conclusion

9-adequate

Reference 

10-adequate

Thank you very much for allowing me to review your manuscript.

Best regards.

Author Response

The manuscript titled "A Qualitative Analysis of Daily Sleep Routines in Adults Not in Employment Living with Type 2 Diabetes Mellitus. " aimed to investigate the extent to which social factors shape sleep timing in retired individuals, or individuals who were not currently working, with T2D.

There are some minor and major considerations that I would like to discuss with the authors.

Abstract

1-Title:Adequate, the authors identify the design and subject matter of the study.

2- Adequate: The abstract gathers the relevant information of the manuscript.

3-It is recommended not to include abbreviations in the abstract without prior explanation.

"Results: Four themes were identified: ”Consistent Sleeping Patterns”, “Fluctuating Sleep Timing”, “Night-Time Disruptions” and “Lasting Effort Needed with T2DM”.

Authors’ Reply: We have now removed the abbreviation, replacing it with “Lasting Effort Needed with Type 2 Diabetes Mellitus”.

4-Change the keywords. Delete the words "Thematic Analysis", "Sleep timing"; "Diabetes"; and "Qualitative". Not found in the MeSH (Medical Subject Headings). It is recommended to put the keywords as they appear in the pubmed thesauri.

Authors’ Reply: We have now amended the keywords according to relevant terms as they appear in the PubMed thesaurus.

Introduction

5-Adequate: The most important concepts of the subject to be developed are identified.

Materials and Methods 

6-Why do the authors incorporate the material and method at the end of the manuscript?

Authors’ Reply: The structure of the manuscript is as laid out in the instructions for authors for this journal.

Results

7-Adequate. The results are correct. The authors present the findings of the study in an organized manner..

Discussion

8-The authors are aware of the limitations of their study design. Could the disparity in disease debuts be a limitation?

Authors’ Reply: We have now included that duration of disease may be a factor that may influence sleep/wake timing: Lines 487-488 “…it is possible that the sleep patterns of patients with serious complication of type 2 DM would vary from those with well controlled or uncomplicated disease or with longstanding disease compared to more recently diagnosed disease”

Conclusion

9-adequate

Reference

10-adequate

Reviewer 3 Report

Comments and Suggestions for Authors

The paper reports a qualitative study on sleep quality in people with DM2. The paper is interesting and easy to understand. It includes new inside on this topic.  However, I have a few comments for the authors:

- your population is not so homogenous. Have you evaluated the influence of these elements in your analyses?

- has this paper suggestions for future studies? Qualitative studies are always helpful for insides from patients.

- when and where the interviews were performed? Was only one person to record this?

Author Response

The paper reports a qualitative study on sleep quality in people with DM2. The paper is interesting and easy to understand. It includes new inside on this topic.  However, I have a few comments for the authors:

- your population is not so homogenous. Have you evaluated the influence of these elements in your analyses?

Author’s Reply: We have not formally examined the role of different disease aspects on sleep/wake habits in the current study. In the “Strengths and Limitations” section of the discussion we do highlight how some of these clinical characteristics might impact on sleep routines: Line 484-488 “Secondly, detailed clinical information was not gathered, and so specific factors such as medical or psychological co-morbidities were not examined; it is possible that the sleep patterns of patients with serious complication of type 2 DM would vary from those with well controlled or uncomplicated disease or with longstanding disease compared to more recently diagnosed disease.”

- has this paper suggestions for future studies? Qualitative studies are always helpful for insides from patients.

Authors’ Reply: We have made a some suggestions for future work in the conclusions section: Lines 597-600 - “We propose that the current findings could be used to guide future quantitative and qualitative research investigating behavioral and psychosocial drivers of sleep timing variability in patients with type 2 DM and inform psychoeducational interventions addressing such sleep timing variability with a view to improving disease management and quality of life.”

- when and where the interviews were performed? Was only one person to record this?

Authors’ Reply: We have now clarified that the first author (RK) conducted all of the interviews and that all interviews were recorded depending on the mode of communication used for the interviews: Lines 523-525 “Interviews were conducted by RK and lasted between 18 and 33 minutes (average duration of 24 minutes) and all interviews took place either online via Microsoft Teams or over the phone, depending on the participants preference and access to technology.”

We also confirm when interviews were conducted: Line 507 “…at the time of the interview (April-September 2021).”

Round 2

Reviewer 2 Report

Comments and Suggestions for Authors

Dear authors and editor,

We would like to thank the authors for their efforts in responding to all the indications. 

Kind regards.